# Contemporary Status of Acute Myocardial Infarction in Korean Patients: Korean Registry of Acute Myocardial Infarction for Regional Cardiocerebrovascular Centers

**DOI:** 10.3390/jcm10030498

**Published:** 2021-02-01

**Authors:** Rock Bum Kim, Jin Yong Hwang, Hyun Woong Park, Ae-Young Her, Jang Hoon Lee, Moo Hyun Kim, Chang Hwan Yoon, Jae Young Cho, Sung-Il Woo, Yongcheol Kim, Jae-Young Han, Joon Hyouk Choi, Song Yi Kim, Si Wan Choi, Sung Ju Jee, Sang Yeub Lee, Ki-Bum Won, Kyeong-Soo Park, Dae Woo Hyun

**Affiliations:** 1Regional Cardiocerebrovascular Disease Center, Gyeongsang National University Hospital, Jinju 52727, Korea; krb747@gmail.com; 2Department of Internal Medicine, Gyeongsang National University School of Medicine and Gyeongsang National University Hospital, Jinju 52727, Korea; chunjium@hanmail.net; 3Division of Cardiology, Department of Internal Medicine, Kangwon National University School of Medicine, Chuncheon 24341, Korea; hermartha@hanmail.net; 4Department of Internal Medicine, Kyungpook National University Hospital, Daegu 41944, Korea; ljhmh75@knu.ac.kr; 5School of Medicine, Kyungpook National University, Daegu 41944, Korea; 6Department of Cardiology, Dong-A University Hospital, Busan 49201, Korea; kimmh@dau.ac.kr; 7Cardiovascular Center, Seoul National University Bundang Hospital, Seoul National University College of Medicine, Seongnam 13620, Korea; changhwanyoon@snubh.org; 8Regional Cardiocerebrovascular Center, Department of Cardiovascular Medicine, Wonkwang University Hospital, Iksan 54538, Korea; librato46@gmail.com; 9Division of Cardiology, Department of Internal Medicine, Inha University Hospital, Incheon 22332, Korea; siwoo@inha.ac.kr; 10Division of Cardiology, Chonnam National University Hospital, Gwangju 61469, Korea; Dr.YongcheolKim@gmail.com; 11Department of Rehabilitation Medicine, Chonnam National University Medical School & Hospital, Gwangju 61469, Korea; white--fish@hanmail.net; 12Division of Cardiology, Department of Internal Medicine, School of Medicine, Jeju National University, Jeju National University Hospital, Jeju 63241, Korea; valgom@naver.com (J.H.C.); ttoromom@jejunu.ac.kr (S.Y.K.); 13Department of Cardiology, Chungnam National University Hospital, Daejeon 35015, Korea; siwanc@cnu.ac.kr; 14Department of Rehabilitation Medicine, Chungnam National University Hospital, Daejeon 35015, Korea; drjeesungju@hamail.net; 15Department of Internal Medicine, Chungbuk National University College of Medicine, Cheongju 28644, Korea; louisahj@gmail.com; 16Division of Cardiology, Ulsan University Hospital, University of Ulsan College of Medicine, Ulsan 44033, Korea; kbwon@uuh.ulsan.kr; 17Regional Cardiocerebrovascular Disease Center, Mokpo Jung-ang Hospital, Mokpo 58615, Korea; kspark386@gmail.com; 18Department of Internal Medicine, Andong General Hospital, Andong 36743, Korea; daewoohyun@naver.com

**Keywords:** acute myocardial infarction, case fatality, registry

## Abstract

Background: This study aimed to present the development process and characteristics of the Korean Registry of Acute Myocardial Infarction for Regional Cardiocerebrovascular Centers (KRAMI-RCC). Methods: We developed KRAMI-RCC, a web-based registry for patients with AMI. Patients from 14 RCCs were registered for more than three years from July 2016. It includes an automatic error-checking system, and user training and on-site monitoring are performed to manage data quality. Results: A total of 11,700 AMI patients were registered in KRAMI-RCC over three years (73.9% men). The proportions of patients with ST-elevation and non-ST-elevation myocardial infarction at discharge were 43.4% and 56.6%, respectively. Of the total three-year patients, 5.6% died in the hospital, and 4.4% died 12 months after discharge. The case fatality within 12 months was 9.7%. Pre-hospital care data showed delayed arrival time after onset of symptoms (median 153 min) and low transportation rate by public ambulance (25.2%). Post-hospital care data showed lower participation rate in the second rehabilitation program (16.8%). Conclusions: The recently developed KRAMI-RCC registry has been more focused on pre-hospital and post-hospital data, which will be helpful in understanding the current state of AMI disease management and in making policy decisions to reduce case fatality in Korea.

## 1. Introduction

Cardiovascular disease (CVD) is the primary cause of death worldwide, accounting for 31.8% of all deaths [1]. It is also a major cause of loss in quality of life, corresponding to 330 million years of life lost and 35.6 million years lived with disability [2]. In particular, ischemic heart disease, represented by acute myocardial infarction (AMI), has the highest mortality rate and causes the poorest quality of life [3].

Therefore, many countries have a registry for AMI patients, through which they perform epidemiological studies, identify risk factors, and conduct policy studies to prevent and reduce mortality. The representative AMI registry is the WHO MONItoring of Trend and Determinants in Cardiovascular Disease registry, which started in 1979, enrolling investigators from 26 countries [4]. The Swedish coronary angiography and angioplasty registry was developed in 1998 [5]. In addition, the National Cardiovascular Data Registry was created in the United States [6], and registries have been developed and operated in several countries such as the United Kingdom [7], Switzerland [8], Canada [9], Italy [10], Singapore [11], Japan [12], and China [13]. In South Korea, the Korea Acute Myocardial Infarction Registry (KAMIR) was developed in 2005 and participates with 41 hospitals [14]. However, KAMIR mainly focuses on clinical information during hospitalization and lacks pre-hospital or post-hospital information including rehabilitation. 

Since 2008, 14 regional cardiocerebrovascular centers (RCCs) have been established and run sequentially by the South Korean government to prevent and reduce mortality in patients with myocardial infarction and stroke, and to close the medical gap between regions. However, the need to develop a registry for prevention, rehabilitation, and policy management as well as research through the RCCs’ AMI patient information emerged. Researchers in the RCCs and government developed the Korean Registry of AMI for RCCs (KRAMI-RCC) in 2015. The main purpose of this registry was to evaluate the regional status of AMI management about pre-hospital, in-hospital, and post-hospital phase, and the performance of each RCC, and finally to improve the regional AMI management system. This study aimed to present the development process and characteristics of the KRAMI-RCC, including the kind of data registered, maintenance of quality, and data characteristics for three years compared with other AMI registries.

## 2. Methods

### 2.1. Participant Hospitals

As of 2020, 14 hospitals have participated in the KRAMI-RCC. At the time of development in 2015, 11 RCCs (Kangwon National University Hospital, Kyungpook National University Hospital, Jeju National University Hospital, Chungbuk National University Hospital, Chonnam National University Hospital, Gyeongsang National University Hospital, Dong-A University Hospital, Wonkwang University Hospital, Chungnam National University Hospital, Seoul National University Bundang Hospital, and Inha University Hospital) participated, and in 2017, Mokpo Jung-ang Hospital and Andong Hospital were included. In 2018, the Ulsan University Hospital was newly established as an RCC and was included in the KRAMI-RCC. 

### 2.2. Requirement for and Initiation of KRAMI-RCC

The main aim of an RCC is to operate a 24 h/7 day treatment system to rapidly treat patients with AMI and stroke and to perform patient education and rehabilitation to prevent recurrence. In addition, it provides a CVD prevention and education campaign for general populations. During this mission, it became necessary to establish integrated data for all AMI patients in RCCs and to build a registry for quality control and evaluation of RCCs. In July 2015, a team consisting of cardiologists, preventive medical physicians, rehabilitation physicians, government stakeholders, and web server developers began developing the KRAMI-RCC. 

### 2.3. Steering Groups and Structure

The steering groups of KRAMI-RCC consist of 19 specialists (15 cardiologists, 2 rehabilitation physicians, and 2 preventive medical physicians) with one or more participating in each RCC. It is divided into four sub-working groups consisting of the indication, research, quality, and web database management teams. The RCC of the Gyeongsang National University Hospital is responsible for the organization. 

The steering groups develop and manage indicators that are enrolled in KRAMI-RCC, report research and statistics using registry data, manage data quality, and train users who input data in the KRAMI-RCC from the hospital records. To this end, steering-group meetings are held at least once a year. Nine offline meetings were held from 2015 to 2019.

### 2.4. Development of a Web-Based Registry and Pilot Test

KRAMI-RCC is a web-based registry in which users of the 14 RCCs enrolled online data of AMI patients from their own hospital records. For this, the website (www.krami.org) was developed in December 2015 in cooperation with an external web server developer. 

After development, the pilot test period was set until June 2016, and the actual patient data from 11 RCCs were entered. Website problems and logical errors of the indicators were checked and corrected. After the pilot test, the entire registration of RCC AMI patient data began on 1 July 2016 (Figure 1).

### 2.5. Registry Patients

Patients registered into the KRAMI-RCC were defined as those diagnosed with ST-elevation myocardial infarction (STEMI) or non-ST-elevation myocardial infarction (NSTEMI) among patients who came through each RCC emergency room. STEMI was defined as a case in which myocardial markers (cardiac troponin) increased with at least one value above the 99th percentile of the normal range and with at least one of the following: Typical myocardial infarction symptoms, elevated ST segment or new-onset left bundle branch block on electrocardiography (ECG), and development of pathological Q waves on ECG. NSTEMI was defined as the same as STEMI, but these ECG findings of STEMI were not found.

From July 2016 to June 2019, 11,894 patients were registered in the KRAMI-RCC. Among them, 136 patients died in the emergency room or were transferred to another hospital prior to diagnosis. Among the remaining 11,758 patients with final diagnosis, 5079 patients had STEMI, 6621 patients had NSTEMI, and 58 had other conditions, such as stress-induced cardiomyopathy or acute myocarditis (Figure 2). The annual number of STEMI patients was 1684 (45.1%), 1683 (42.8%), and 1712 (42.4%), and the number of NSTEMI patients was 2053 (54.9%), 2247 (57.2%), and 2321 (57.6%) (Figure 3).

### 2.6. Data Collection: Input Indicators

The total number of input indicators of KRAMI-RCC was 392, which was divided into three major categories: (1) Pre-hospital data, (2) hospital data, and (3) post-hospital data. They were further divided into the following 17 subcategories: (1) Patient information, (2) information before arriving at the hospital, (3) information on emergency medical services, (4) visit to another hospital, (5) emergency room information, (6) coronary angiography room information, (7) past medical history, (8) event during admission, (9) laboratory findings, (10) medication at discharge, (11) education during admission, (12) information on cardiac rehabilitation, (13) status at discharge, (14) outpatient education, (15) information at 3 months and 12 months after discharge, (16) outpatient cardiac rehabilitation, and (17) event for outcomes. Detailed input indicators are attached as Appendix A.

### 2.7. Data Quality

To maintain registry quality, the registry has been programmed to automatically display colors as signals when there are errors in the input data (missing, mistyping, out of range, etc.). In other words, when a data error occurs, it is marked with red light, and when the error is corrected, the red light changes to green. A data error check is performed by one dedicated researcher every day, and if an error occurs, the input user of the RCC is contacted via email or phone to check and correct the error.

Approximately 30 users register patient data from the 14 RCCs to the KRAMI-RCC. They are trained on registry input, error prevention, and data status at least once a year. In addition, the dedicated researcher conducts an audit using 10% randomized web input data and actual hospital data through 14 RCC visits once a year. If there are errors between the web data and hospital data, the web data are corrected after rechecking. The user who made the error was then trained once again.

### 2.8. Reporting and Use of Data

Since 2017, KRAMI-RCC has published an annual report on the basic characteristics of patients and key indicator results. The 2017 report analyzed the patients entered between 1 July 2016 and 31 October 2017. The outcomes such as death, drug adherence, complications, etc., of registered AMI patient is monitored by telephone at 30 days and 1 year after discharge. The telephone number of patients and their next of kin was not registered at KRAMI-RCC, but identified at each electronic medical record of RCC after obtaining consent. The follow-up results of the 2017 report were analyzed only in patients who underwent final monitoring.

The 2018 report analyzed the patients registered from 1 July 2016 to 31 December 2017, while the 2019 report analyzed data from 1 July 2016 to 30 June 2019. The annual reports are uploaded to the KRAMI-RCC website, after which they are available for anyone to download and read. 

Researchers in the steering groups can analyze the data registered at their hospital. However, to analyze the data of all 14 RCCs, a study proposal is submitted to the research management team for review. If there is no issue in the study proposal, the researcher will receive the full data all 14 RCCs. After the analysis, before submitting the research results, the research management team is reviewed once again, and if there are no problems, the final publication will be made.

### 2.9. Comparison of KRAMI-RCC with Other AMI Registries

Recently published papers within approximately 10 years were searched, and the main characteristics of KRAMI-RCC, such as mean or median age, proportion of male patients, STEMI, risk factors, and mortality, were compared. 

The following seven AMI registries were compared: Korea Acute Myocardial Infarction Registry-National Institutes of Health (KAMIR-NIH) [15], Japanese Acute Myocardial Infarction Registry (JAMIR) [16], China Acute Myocardial Infarction (CAMI) Registry [13], Swedish Web-System for Enhancement and Development of Evidence-Based Care in Heart Disease Evaluated According to Recommended Therapies/Register of Information and Knowledge About Swedish Heart Intensive Care Admissions (SWEDEHEART/RIKS-HIA) [17], Acute Myocardial Infarction in Switzerland (AMIS) Plus [8,18], Myocardial Ischemia National Audit Project (MINAP) [7], and Acute Coronary Treatment and Intervention Outcomes Network Registry–Get With the Guidelines (ACTION Registry-GWTG) [19]. 

### 2.10. Ethical Issue 

No personal information (patient’s name, address, ID, phone number, hospital ID) was registered in KRAMI-RCC. Initially, all RCC registries were deliberated and approved depending on the individual hospital IRB situation. Although KRAMI-RCC was a public cardiovascular statistic project and has an exemption for private information protection according to the Act on the Prevention and Management of the Cardiocerebrovascular Disease No. 6, data collection and management were performed under private information protection law.

## 3. Results

### 3.1. Demographic Features and Risk Factors

A total of 11,700 AMI patients were registered in the KRAMI-RCC for 3 years (1 July 2016, to 30 June 2019). The mean age was 65.9 (standard deviation (SD) ± 13.0) years, and the proportion of patients aged 55–64 was the largest (25.5%).

Of these patients, 8652 (73.9%) were male, the mean age was 63.0 (SD ± 12.4) years, and the proportion of patients aged 55–64 was the largest (29.8%). There were 3048 (26.1%) female patients, with an average age of 74.0 (SD ± 11.0) years, and 39.8% of the female patients were 75–84 years old (Figure 4).

Most of the patients were public insurance policyholders (92.7%), those with a lower education level (below high school) accounted for up to 71.0%, and the rate of single-living pattern was 19.1%. Predominant risk factors were hypertension (51.1%), current smoking (36.3%), diabetes mellitus (30.0%), excessive alcohol use (13.3%), and dyslipidemia (12.1%). Interestingly, 13.8% of the patients had undergone percutaneous coronary intervention (PCI), which might be an important risk factor (Table 1).

### 3.2. Pre-Hospital Data

Moreover, 62.8% of the patients had initial symptoms of AMI at home, and 26.1% had AMI onset outdoors. The most common symptom was chest pain (87.5%). In 16.4% of the patients, AMI was recognized at the onset of symptoms. Among STEMI patients, the symptom-to-first medical contact, symptom-to-arrival, and door-to-device median times were 66.0 (interquartile range [IQR] 30.0–180.0), 119.0 (IQR 60.0–240.0), and 55.0 (IQR 44.0–66.0) min, respectively. The proportion of patients who arrived within 60 min after symptom onset was only 23.0%. As regards transportation, 31.4% of STEMI patients were transported by public ambulance (Table 2). 

### 3.3. In-Hospital Data

Only 37.3% of the STEMI patients were treated by a cardiologist at a regular work time. The rest of the patients were treated outside the regular work time, which means that a 24/7 duty system is mandatory for managing STEMI patients. Most of the AMI patients underwent PCI (85.6% and 97.2% of NSTEMI and STEMI patients, respectively). One-vessel disease was the most common (59.4%), and the left anterior descending artery was the most often involved (48.5%).

Serious complications from AMI at the emergency department were as follows: Heart failure (8.9%), shock (7.8%), cardiac arrest at hospital (6.4%), stroke at hospital (0.8%), and atrial fibrillation at hospital stay (8.3%). 

Well-trained coordinators provided patient education for secondary prevention, which was accomplished in the form of disease information (90.5% of all patients), medication information and importance of compliance (90.4% of all patients), smoking abstinence (85.7% of current smokers), and cardiac rehabilitation (85.6% of all patients). In addition, 75.7% of all patients participated in the first rehabilitation program during admission.

Along with aspirin, the most often used P2Y12 antagonists were clopidogrel (55.6%), ticagrelor (33.1%), and prasugrel (4.9%). Most patients received statins (92.4%), and combined lipid-lowering therapy with statin and ezetimibe was prescribed for only 8.1% of all patients at discharge (Table 3).

### 3.4. Post-Hospital Data and Mortality

Most AMI patients were successfully monitored until 3 and 12 months after discharge (89.7% and 84.5%, respectively). The drug adherence rate was very high until 12 months after discharge. The rates of drug adherence at 3 months and 12 months after discharge were 99.4% and 99.6%, respectively. Most AMI patients returned to work or daily life until 3 months after discharge (94.4%). However, the participation rate in the second rehabilitation after discharge was lower (i.e., 14.4% in NSTEMI patients and 19.8% in STEMI patients).

KRAMI-RCC monitors patient death or other events by telephone at 3 and 12 months after discharge. Of the 11,700 patients in 3 years, 655 (5.6%) died in the hospital and 11,045 (94.4%) were discharged. Among the discharged patients, 205 died at 3 months, and the accumulated mortality rate was 7.4%, including hospital deaths. The total number of patients who died until 12 months after hospital admission was 1139 (9.7%). The in-hospital, 3-month, and 12-month mortality rates for each year are plotted in Figure 5.

During the 12 months after discharge, the all-cause death rate was 4.4% (484/11,045) among the discharged AMI patients, and the proportion of cardiovascular deaths was 34.3% (166/484). Among the surviving patients, major complications with lower rates were cerebrovascular events (0.8%), heart failure (1.8%), and Bleeding Academic Research Consortium >2 (1.3%) (Table 4).

### 3.5. Difference between Genders

There were several differences in the characteristics of registered AMI patients according to gender. In males, the STEMI proportion (46.0%) was higher than that in females (35.9%), and the mean age (male 63.0 years, female 74.0 years) was lower. There were also differences in clinical characteristics such as the proportions of some risk factors and the proportion of PCI. In-hospital mortality (male 4.8%, female 7.9%), 3-month (male 1.5%, female 2.9%), and 1-year (male 3.7%, female 6.6%) mortality were all higher in women (Appendix A).

## 4. Discussion

Herein, we present the purpose and development process of the KRAMI-RCC registry. We also reported demographic, pre-hospital, in-hospital, and post-hospital data and outcome data of approximately 11,700 patients registered in the KRAMI-RCC over the past 3 years. 

In South Korea, a registry for some diseases (especially cancer [20], rare diseases [21,22], or heart failure [23]) has been developed; however, registries for other diseases remain insufficient. Another AMI patient registry in South Korea is KAMIR [14]. KAMIR was started in 2005 with 55 participating hospitals and has been in operation to the present. However, KAMIR has been developed and run by researchers or academic societies. Thus, it is primarily intended for disease research and not for policymaking and national project evaluation.

KRAMI-RCC is a registry developed, run, and supported by the state since 2015 along with RCC support projects in 2009. Therefore, it aims to create evidence in making policy decisions with the evaluation of RCCs in addition to clinical studies. This purpose is different from that of KAMIR. In addition, the registration data of KRAMI-RCC differ from those of KAMIR in that it contains data on cardiac rehabilitation and patient education as well as data on socioeconomic factors and coronary risk factors.

In this study, the 12-month mortality rate, including in-hospital mortality, was 9.7%, which was lower than the 13.1% of a recent analysis using Korean health insurance claims data [24]. The claims data revealed a higher mortality rate because it included patients admitted in secondary hospitals that were not treated 24/7 or lacked services for secondary prevention. By contrast, 14 RCCs maintain a 24/7 system for rapid treatment and provide patient education and cardiac rehabilitation during hospitalization. Therefore, to lower the AMI mortality rate in Korea, it is necessary to expand the care system, that is, the patient receives continuous treatment during transfer to another hospital. Interestingly, although this mortality data, which were used to present a general trend, was actually obtained within a short period, the post-hospital mortality rate markedly decreased compared to the in-hospital mortality rate (4.7%, 4.2%, and 3.5% over 3 years). It might be associated with increasing hospital patient education by trained coordinators over 3 years, resulting in excellent drug compliance and a high success rate of abstaining from smoking. Although only 28% of the patients participated in the second rehabilitation program after discharge, increased participation in rehabilitation programs may have contributed to the decrease in post-hospital mortality rate. However, these results obtained by education and advertisement were limited and less cost effective when used to improve and recognize AMI symptoms prior to arrival to a PCI-capable hospital. Thus, to reduce the hospital mortality rate in Korea, strategies that focused on secondary prevention may be cost effective in achieving better survival.

In-hospital patient education for drug compliance and smoking cessation should be recommended to all AMI hospitals, and AMI rehabilitation programs and hospital-based, home-based, or community-based strategies should be encouraged by physicians and supported by medical policy to reduce AMI mortality.

AMI registries have individual and specific aims, different data collection systems, and different populations; thus, it is not possible to compare registries using the head-to-head approach (Table 5). In the last decade, in-hospital mortality and 12-month mortality varied depending on specific national registries. For example, the following in-hospital mortalities were similar: 5.6% in KRAMI-RCC, 3.9% in KAMIR-NIH, and 4.5% in CAMI Registry. However, the ACTION registry showed an exceptionally lower in-hospital mortality rate (2.8%), and the JAMIR and SWEDEHEART registries showed an exceptionally higher in-hospital mortality rate (8.3% and 7.2%, respectively), which might be associated with older mean age. KRAMI-RCC and KAMIR-NIH showed a decreasing proportion of STEMI and increasing proportion of NSTEMI patients, similar to SWEDE-HEART, MINAP, and ACTION registries, but contrary to AMIS-PLUS, that includes the data from 1997 to 2009. Among coronary risk factors, hypertension, smoking, and diabetes mellitus were the most common factors in all registries. However, dyslipidemia showed a relatively lower proportion among risk factors in Korean and Chinese registries than in western and Japanese registries. This result suggests that high-risk patients were not treated for dyslipidemia because KRAMI-RCC defined dyslipidemia as patients taking dyslipidemia medications or recommended medication from physician.

To interpret the data from KRAMI-RCC, its limitations should be noted. First, the KRAMI-RCC registry does not include all AMI patients in South Korea. In particular, the KRAMI-RCC did not cover AMI patients living in Seoul, which account for approximately 20% of the Korean population. Therefore, these data might only be representative of all Korean patients in areas other than Seoul. Furthermore, only one center was designated in larger cities such as Busan, Deagu, Deajeon, and Inchean; thus, the center registers and reflects data of numerous AMI patients from several hospitals in these cities. Some RCC data covered only data of a small city, and some RCCs were designated to cover an entire province. In fact, KRAMI-RCC patients account for approximately 16% of all AMI patients in Korea, as obtained from the health insurance corporation claims data. However, RCCs were evenly distributed in areas other than Seoul, which might provide representative data.

Second, all RCCs were tertiary hospitals with qualified critical pathways, facilities, and specialized personnel, including coordinator nurses dedicated to the education of AMI patients and cardiologists with a 24/7 duty system. Therefore, this result should not be extrapolated to secondary hospitals or clinics without supporting facilities and personnel. 

Third, the coordinators of the KRAMI-RCC monitored the 3-month and 12-month outcomes of patients by telephone after discharge. If the patient or caregiver cannot be contacted by telephone or cannot give clear responses, post-hospital data may be underestimated or overestimated. To solve these issues, the coordinator has been trained about the follow-up manual, and some RCCs use complimentary data such as death statistics, health insurance corporation data, and outpatient clinic monitoring. To overcome these limitations, all data of KRAMI-RCC should be linked with health insurance corporation data in the future. 

## 5. Conclusions

Four years have passed since KRAMI-RCC was developed and RCC AMI patient registration was started. Since then, the registry has provided consistent data quality. KRAMA-RCC has been more focused on pre-hospital and post-hospital data, which will be helpful in understanding the current state of AMI disease management and making policy decisions to reduce AMI case fatality in Korea. KRAMA-RCC is also expected to provide important information for AMI disease research through continuous support and expansion in the future.

## Figures and Tables

**Figure 1 jcm-10-00498-f001:**
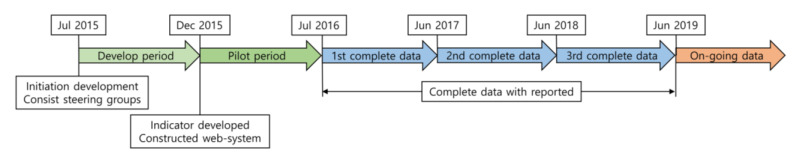
Development process and registered data over time period of the Korean Registry of Acute Myocardial Infarction for Regional Cardiocerebrovascular Centers (KRAMI-RCC).

**Figure 2 jcm-10-00498-f002:**
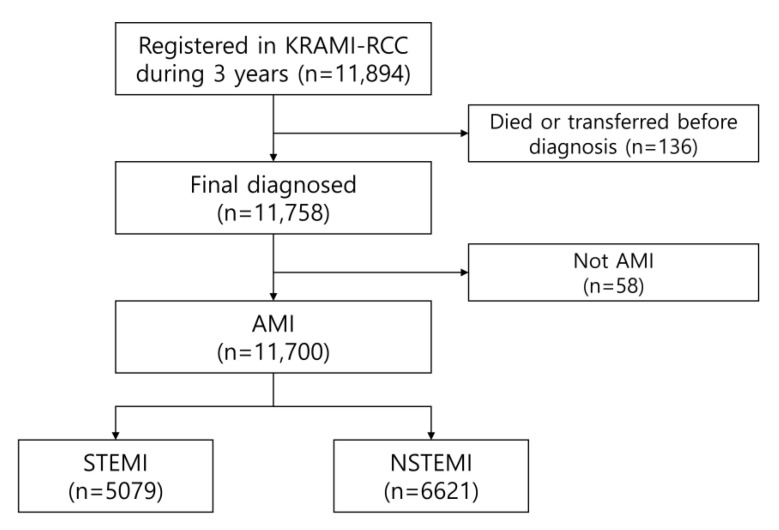
The number of AMI patients for analysis registered in the KRAMI-RCC during 3 years.

**Figure 3 jcm-10-00498-f003:**
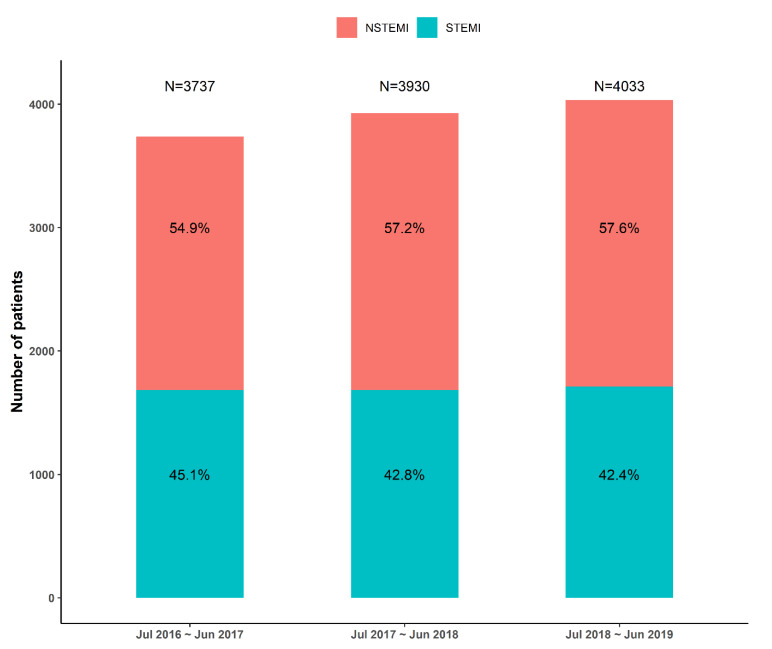
Proportions of AMI type of KRAMI-RCC each 3 years. NSTEMI: non-ST-elevation myocardial infarction.

**Figure 4 jcm-10-00498-f004:**
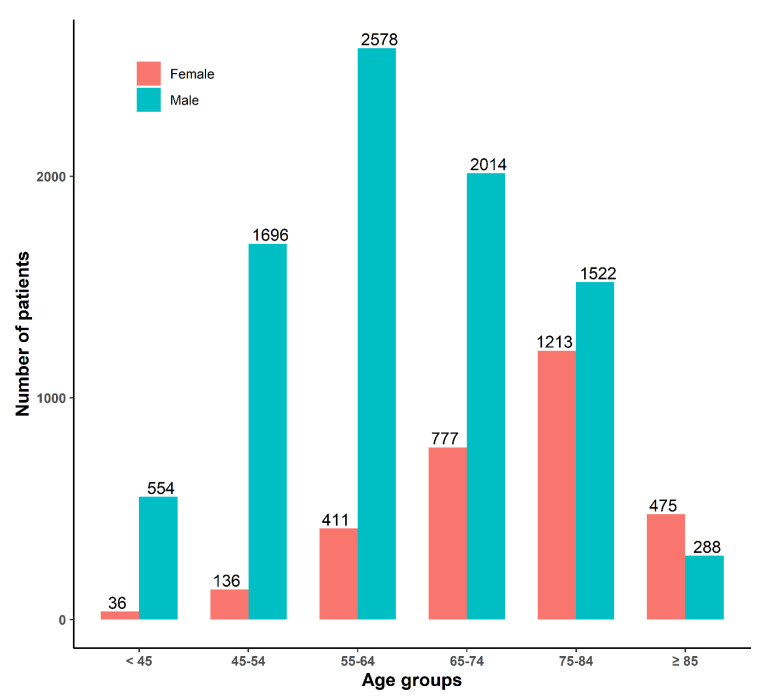
Number of patients by gender and by age groups on the KRAMI-RCC during 3 years.

**Figure 5 jcm-10-00498-f005:**
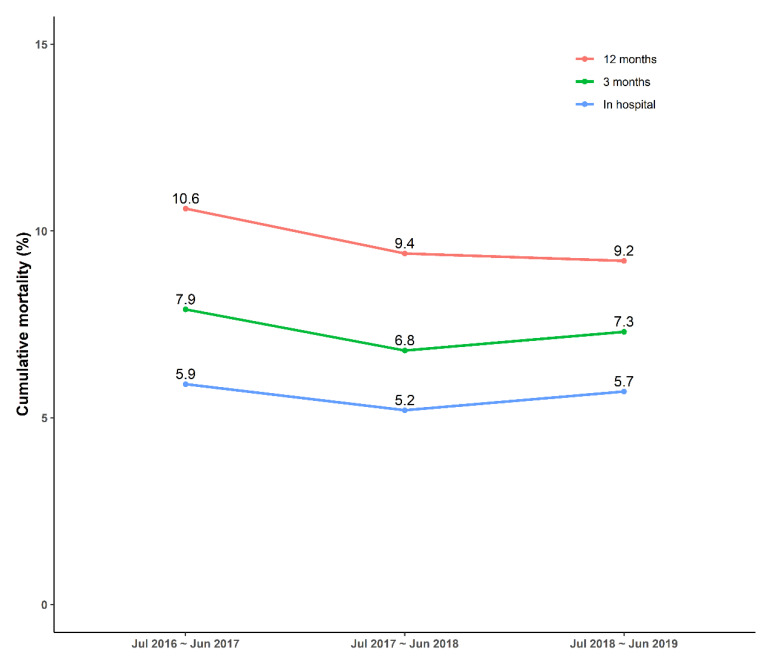
Cumulative mortality of in-hospital, 3 months, and 12 months after admission on the KRAMI-RCC each 3 years.

**Table 1 jcm-10-00498-t001:** Demographic characteristics of the KRAMI-RCC patients by AMI type at arrival.

Characteristics	NSTEMI	STEMI	Total
	(*N* = 6621)	(*N* = 5079)	(*N* = 11,700) ^a^
Age (years), mean ± SD	67.2 ± 12.7	64.1 ± 13.2	65.9 ± 13.0
Male, *N* (%)	4668 (70.5%)	3984 (78.4%)	8652 (73.9%)
Body mass index (kg/m^2^), mean ± SD	24.0 ± 3.4	24.1 ± 3.3	24.0 ± 3.4
Type of health insurance			
Public insurance	6125 (92.6%)	4719 (93.0%)	10,844 (92.7%)
Medical aid	449 (6.8%)	307 (6.0%)	756 (6.5%)
Individual insurance	43 (0.6%)	49 (1.0%)	92 (0.8%)
Education level			
Elementary school	1749 (26.4%)	1173 (23.1%)	2922 (25.0%)
Middle school	966 (14.6%)	667 (13.1%)	1633 (14.0%)
High school	2046 (30.9%)	1700 (33.5%)	3746 (32.0%)
College	1215 (18.4%)	1052 (20.7%)	2267 (19.4%)
Graduate school	182 (2.7%)	147 (2.9%)	329 (2.8%)
Unknown	463 (7.0%)	338 (6.7%)	801 (6.8%)
Living situation			
Alone	1268 (19.2%)	966 (19.0%)	2234 (19.1%)
With family	5239 (79.1%)	4011 (79.0%)	9250 (79.1%)
Cohabitation	114 (1.7%)	101 (2.0%)	215 (1.8%)
Risk factors and Medical history			
Current smoking	2077 (31.4%)	2164 (42.6%)	4241 (36.3%)
High-risk alcohol use ^b^	813 (12.3%)	739 (14.6%)	1552 (13.3%)
Hypertension	3674 (55.5%)	2308 (45.5%)	5982 (51.1%)
Dyslipidemia	920 (13.9%)	497 (9.8%)	1417 (12.1%)
Diabetes Mellitus	2240 (33.8%)	1274 (25.1%)	3514 (30.0%)
Previous PCI	1166 (17.6%)	448 (8.8%)	1614 (13.8%)
Previous CABG	94 (1.4%)	15 (0.3%)	109 (0.9%)
Previous dialysis	253 (3.8%)	44 (0.9%)	297 (2.5%)
Any cancer	372 (5.6%)	248 (4.9%)	620 (5.3%)

NSTEMI; non-ST elevation myocardial infarction, STEMI; ST elevation myocardial infarction, PCI; percutaneous coronary intervention, CABG; coronary artery bypass graft, SD; standard deviation. ^a^ Among registered 11,894 patients, 136 were excluded because they died or were transferred to another hospital prior to diagnosis. ^b^ More than 3 drinks per week and more than 5 shot glasses per one drink for men, more than 3 shot glasses per one drink for women.

**Table 2 jcm-10-00498-t002:** Pre-hospital clinical characteristics of the KRAMI-RCC patients by diagnosis of NSTEMI and STEMI at arrival.

Characteristics	NSTEMI (*N* = 6621)	STEMI (*N* = 5079)	Total (*N* = 11,700) ^a^
Symptom occurring places			
Unknown	821 (12.4%)	472 (9.3%)	1293 (11.1%)
In home	4215 (63.7%)	3122 (61.6%)	7337 (62.8%)
Outdoors	1576 (23.8%)	1476 (29.1%)	3052 (26.1%)
Symptom type			
Chest pain	5603 (84.6%)	4634 (91.2%)	10,237 (87.5%)
Dyspnea	2251 (34.0%)	1330 (26.2%)	3581 (30.6%)
Others ^b^	3766 (56.9%)	3151 (62.0%)	6917 (59.1%)
Cognition of AMI at symptom	1222 (18.5%)	699 (13.8%)	1921 (16.4%)
Symptom to first medical contact time (min), median (IQR)	120.0 (50.0;343.0)	66.0 (30.0;180.0)	90.0 (37.0;248.0)
Symptom to arrival time (min), median (IQR)	196.0 (85.0;452.0)	119.0 (60.0;240.0)	153.0 (65.0;336.0)
Symptom to arrival time			
<=60 min	915 (18.5%)	1291 (27.8%)	2206 (23.0%)
60–180 min	1425 (28.8%)	1746 (37.6%)	3171 (33.0%)
>180 min	2614 (52.8%)	1607 (34.6%)	4221 (44.0%)
Door to device time at STEMI patients (min), median (IQR)	-	55.0 (44.0;66.0)	-
Type of transport to emergency room			
Public ambulance	1354 (20.5%)	1596 (31.4%)	2950 (25.2%)
Private ambulance	2293 (34.6%)	2132 (42.0%)	4425 (37.8%)
Cares or buses	2887 (43.6%)	1327 (26.1%)	4214 (36.0%)
Walking	87 (1.3%)	24 (0.5%)	111 (0.9%)
Transfer from other hospitals	3330 (50.3%)	2670 (52.6%)	6000 (51.3%)

NSTEMI: Non-ST elevation myocardial infarction; STEMI: ST elevation myocardial infarction; SD: Standard deviation. ^a^ Among registered 11,894 patients, 136 were excluded because they died or were transferred to another hospital prior to diagnosis. ^b^ One or more of the following symptoms: Loss of consciousness, cold sweat, systemic weakness, vertigo, stomach pain, and radiating pain.

**Table 3 jcm-10-00498-t003:** In-hospital clinical characteristics of the KRAMI-RCC patients by diagnosis of NSTEMI and STEMI at arrival.

Characteristics	NSTEMI (*N* = 6621)	STEMI (*N* = 5079)	Total (*N* = 11,700) ^a^
Arrival by working time of specialists			
Day time on duty	3968 (70.0%)	1839 (37.3%)	5807 (54.8%)
Night time on duty	1518 (26.8%)	2524 (51.1%)	4042 (38.1%)
On call duty without hospital stay	183 (3.2%)	573 (11.6%)	756 (7.1%)
Performed PCI	5669 (85.6%)	4936 (97.2%)	10,605 (90.6%)
Number of significant stenotic artery			
1	3154 (55.6%)	3150 (63.8%)	6304 (59.4%)
2	1751 (30.9%)	1268 (25.7%)	3019 (28.5%)
3	764 (13.5%)	518 (10.5%)	1282 (12.1%)
Location of stenotic artery			
Left anterior descending	2610 (46.0%)	2529 (51.2%)	5139 (48.5%)
Right coronary artery	1477 (26.1%)	1854 (37.6%)	3331 (31.4%)
Left circumflex	1268 (22.4%)	445 (9.0%)	1713 (16.2%)
Left main	183 (3.2%)	91 (1.8%)	274 (2.6%)
Unknown	131 (2.3%)	17 (0.3%)	148 (1.4%)
Heart failure at arrival	650 (9.8%)	392 (7.7%)	1042 (8.9%)
Cardiogenic shock at arrival	324 (4.9%)	594 (11.7%)	918 (7.8%)
Cardiac arrest at hospital	268 (4.0%)	483 (9.5%)	751 (6.4%)
Stroke at hospital	49 (0.7%)	44 (0.9%)	93 (0.8%)
Atrial fibrillation at hospital	526 (7.9%)	440 (8.7%)	966 (8.3%)
In-hospital death	255 (3.9%)	400 (7.9%)	655 (5.6%)
Education about AMI	6025 (91.1%)	4547 (89.7%)	10,572 (90.5%)
Education about medication	6024 (91.1%)	4542 (89.6%)	10,566 (90.4%)
Education of abstaining smoking among the smokers			
No	302 (14.5%)	303 (14.0%)	605 (14.3%)
Yes	1774 (85.5%)	1860 (86.0%)	3634 (85.7%)
Education about cardiac rehabilitation	5470 (85.8%)	4279 (85.4%)	9749 (85.6%)
Participation in first rehabilitation program	4064 (73.2%)	3374 (79.0%)	7438 (75.7%)
Medication at discharge			
Aspirin	6010 (94.2%)	4567 (97.1%)	10,577 (95.4%)
Clopidogrel	4041 (63.3%)	2119 (45.1%)	6160 (55.6%)
Prasugrel	228 (3.6%)	313 (6.7%)	541 (4.9%)
Ticagrelor	1570 (24.6%)	2098 (44.6%)	3668 (33.1%)
Cilostazol	189 (3.0%)	134 (2.8%)	323 (2.9%)
β-blocker	4884 (76.5%)	3944 (83.9%)	8828 (79.6%)
Calcium channel blocker	1034 (16.2%)	236 (5.0%)	1270 (11.5%)
Angiotensin-converting enzyme inhibitors	2067 (32.4%)	1851 (39.4%)	3918 (35.3%)
Angiotensin II receptor blockers	2369 (37.1%)	1748 (37.2%)	4117 (37.1%)
Statin	5831 (91.4%)	4410 (93.8%)	10,241 (92.4%)
Fibrate	44 (0.7%)	26 (0.6%)	70 (0.6%)
Ezetimibe	484 (7.6%)	417 (8.9%)	901 (8.1%)
Warfarin	108 (1.7%)	83 (1.8%)	191 (1.7%)
New oral anticoagulation	268 (4.2%)	184 (3.9%)	452 (4.1%)
Oral anti-diabetic drug	1517 (23.8%)	943 (20.1%)	2460 (22.2%)
Insulin treatment	222 (3.5%)	132 (2.8%)	354 (3.2%)

NSTEMI: Non-ST elevation myocardial infarction; STEMI: ST elevation myocardial infarction; PCI: Percutaneous Coronary Intervention. ^a^ Among registered 11,894 patients, 136 were excluded because they died or were transferred to another hospital prior to diagnosis.

**Table 4 jcm-10-00498-t004:** Post-hospital clinical characteristics of the KRAMI-RCC patients by diagnosis of NSTEMI and STEMI at arrival.

Characteristics	NSTEMI (*N* = 6366)	STEMI (*N* = 4679)	Total (*N* = 11,045) ^a^
Success of monitoring at PH 3 months	5608 (89.1%)	4155 (90.6%)	9763 (89.7%)
Drug adherence at PH 3 months ^b^			
Low adherence (score >= 4)	32 (0.6%)	26 (0.6%)	58 (0.6%)
High adherence (score < 4)	5459 (99.4%)	4037 (99.4%)	9496 (99.4%)
Participation in second rehabilitation	831 (14.4%)	860 (19.8%)	1691 (16.8%)
No return to work or daily life until PH 3 months	360 (6.4%)	185 (4.5%)	545 (5.6%)
Success of monitoring at PH 12 months	4749 (82.3%)	3711 (87.6%)	8460 (84.5%)
Drug adherence at PH 12 months ^b^			
Low adherence (score >= 4)	16 (0.3%)	17 (0.5%)	33 (0.4%)
High adherence (score < 4)	4646 (99.7%)	3629 (99.5%)	8275 (99.6%)
All cause death from discharge during PH 3 months	124 (1.9%)	81 (1.7%)	205 (1.9%)
All cause death from discharge during PH 12 months	326 (5.1%)	158 (3.4%)	484 (4.4%)
Type of death			
Cardiovascular	114 (35.0%)	52 (32.9%)	166 (34.3%)
Non-cardiovascular	111 (34.0%)	52 (32.9%)	163 (33.7%)
Unknown	101 (31.0%)	54 (34.2%)	155 (32.0%)
AMI re-event during PH 12 month	174 (2.9%)	75 (1.7%)	249 (2.4%)
Cerebrovascular event during PH 12 month	60 (1.0%)	19 (0.4%)	79 (0.8%)
Heart failure event during PH 12 month	116 (1.9%)	69 (1.6%)	185 (1.8%)
Above BARC 2 Bleeding event during PH 12 month	88 (1.5%)	51 (1.2%)	139 (1.3%)

NSTEMI: Non-ST elevation myocardial infarction; STEMI: ST elevation myocardial infarction; PH: Post Hospital, BARC: Bleeding Academic Research Consortium. ^a^ Among registered 11,894 patients, 136 were excluded because they died or were transferred to another hospital prior to diagnosis, and 660 were excluded because they died at hospital. ^b^ Results by Morisky Medication Adherence Scale, which consists of 6 questions (yes/no) items and so scores range 0 to 6.

**Table 5 jcm-10-00498-t005:** Comparison of main characteristics with other AMI registries.

	KRAMI-RCC	KAMIR-NIH [15]	JAMIR [16]	CAMI Registry [13]	SWEDE-HEART [17]	AMIS-PLUS [18]	MINAP [7]	ACTION-Registry [19]
Country	South Korea	South Korea	Japan	China	Sweden	Swiss	UK	USA
Time of data	Jul. 2016–Jun. 2019	Nov. 2011–Oct. 2015	Jan. 2011–Dec. 2013	Jan. 2013–Sep. 2014	Jan. 2006–Dec. 2013	Jan. 1997–Jun. 2009	Jan.2012–Jun. 2013	Jan.2007–Dec. 2014
Participating hospital	14 hospitals	20 hospitals	10 registries network	108 hospitals	74 hospitals	76 hospitals	220	>750 hospitals
Number of patients	11,758	13,624	20,462	26,103	289,699	31,010	118,075	322,523
Age, years	65.9 (SD: 13.0)	64.1	68.8 (SD: 13.3)	63.0 (IQR: 53.0–72.0)	72.0 (IQR: 63.0–81.0)	65.6 (SD: 13.2)	68.5 (SD: 14.0)	61.0 (IQR: 52.0–71.0)
Male,%	74.0	73.5	74.7	74.0	63.7	72.4	66.8	65.1
STEMI,%	43.4	48.2	79.7	-	34.0	58.1	40.1	45.0
Coronary Risk factor,%								
Hypertension	51.1	51.2	63.6	52.0	50.0	57.9	50.6	65.0
DM	30.0	28.6	32.8	20.0	24.2	20.0	21.2	26.0
Dyslipidemia	12.1	11.2	46.2	8.0		57.0	34.2	50.0
Current Smoking	36.2	38.5	34.5	44.0	20.1	38.1	61.4 (current or ex-smoker)	38.0
Previous MI	14.8	8.1	-	-	28.5	39.1 (previous CAD)	34.2	6.0
In hospital mortality,%	5.6	3.9	8.3	4.9 (STEMI)/4.2 (NSTEMI)	7.2	6.8	-	2.8
12 month mortality after discharge,%	4.4	4.3	-	-	12.0	4.0	-	-

NSTEMI: Non-ST elevation myocardial infarction; STEMI: ST elevation myocardial infarction; DM: Diabetes Mellitus; CAD: Coronary Artery Disease.

## Data Availability

The data presented in this study are available on request from the KRAMI-RCC Steering Committee after a deliberation process. The availability of these data is restricted to participating researchers and is therefore not available to the public.

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
