# Peer review of "Contemporary Status of Acute Myocardial Infarction in Korean Patients: Korean Registry of Acute Myocardial Infarction for Regional Cardiocerebrovascular Centers"

_jcm, 2021, doi:10.3390/jcm10030498_

Round 1

Reviewer 1 Report

My only question is:

How was it posssible to follow-up the patients (mortality, drug adherence, complications, etc.) if no personal information was not registered?

This isssue should be clarified in the methods section.

Author Response

Response)

The KRAMI-RCC registration system does not collect personal information, but each regional center collect personal information such as the phone number of each patient and their next of kin for the education and outcome follow up with consent. Researchers at each hospital used the telephone to check the patient's death, drug use, and complications at 30 days and 1 year after discharge. Final follow up data with deidentification were registered on the KRAMI-RCC.

These contents are described in more detail in the line 176-179 of the revised manuscript.

Reviewer 2 Report

Very valuable data-set, showing everyday clinical practice in South Korean acute MI network.

I guess this was not the primary goal of the current publication, but could the Authors give a number of acute MI since the beginning of COVID pandemics? Was there any reduction of MI hospitalizations and transfer Times in 2020 vs. 2019?

Findings:

There is a striking difference in male vs. female patients. I think it deserves some exploratory analysis because of the social impact of this fact. Were there any differences in terms of time to presentation, non-occlusive disease? Risk factors between the groups? Also, was there any difference in the use of PCI in females and males who were admitted with AMI?

Was there a difference in the rate of STEMI/NSTEMI in males and females and a difference in mortality?

Discussion:

Main finding no 3: „AMI patients had a lower level of 319 education and a higher proportion of living alone”. Please specify in relation to what group?

Author Response

Very valuable data-set, showing everyday clinical practice in South Korean acute MI network.

I guess this was not the primary goal of the current publication, but could the Authors give a number of acute MI since the beginning of COVID pandemics? Was there any reduction of MI hospitalizations and transfer Times in 2020 vs. 2019?

 Response)

As mentioned in method (line 139), we analyzed the patients who only arrived at the hospital during Jul. 2016 ~ Jun. 2019. Therefore, we can't compare the monthly number of arrived patients between 2020 and 2019. We would appreciate your understanding of this point. In the future, we planned to check whether the number of AMI patients during the COVID-19.

Findings:

 There is a striking difference in male vs. female patients. I think it deserves some exploratory analysis because of the social impact of this fact. Were there any differences in terms of time to presentation, non-occlusive disease? Risk factors between the groups? Also, was there any difference in the use of PCI in females and males who were admitted with AMI?

Was there a difference in the rate of STEMI/NSTEMI in males and females and a difference in mortality?

 Response)

As your comments, there is a difference in the characteristics of AMI between males and females. So we further analyzed the gender differences and presented the results. The results were described at line 292 - 298 of the revised manuscript. And the table is presented as a supplementary table 1.

Discussion:

Main finding no 3: „AMI patients had a lower level of 319 education and a higher proportion of living alone”. Please specify in relation to what group?

 Response)

The sentence for the main finding on the discussion as you pointed, we deleted the sentence for brevity and clarity and to avoid conflict with another reviewer comment.

Reviewer 3 Report

The authors present a description of a Korean registry of AMI patients conducted in several tertiary centers in Korea in the past 4 years. The main focus was on the different stages of care along with examination of policy changes. There is no doubt that the collection of data and analyzing it, is an important feature of any policy design and assessment of its impact.

I have several comments:

  • The manuscript can be shortened as there are many technical issues that could be left out. In addition, repetition of the results in the first paragraph of the discussion is not required. Moreover, the comparison with other registries should be moved to the discussion.
  • The main purpose of the registry is missing as there are no analysis of policies which are the main goal of data collection in this registry. As such I suggest either to describe AMI patients in the registry or show some implementation of policy changes

Author Response

The authors present a description of a Korean registry of AMI patients conducted in several tertiary centers in Korea in the past 4 years. The main focus was on the different stages of care along with examination of policy changes. There is no doubt that the collection of data and analyzing it, is an important feature of any policy design and assessment of its impact.

I have several comments:

  • The manuscript can be shortened as there are many technical issues that could be left out. In addition, repetition of the results in the first paragraph of the discussion is not required. Moreover, the comparison with other registries should be moved to the discussion.

Response)

As your comment, we removed the main findings sentences in the discussion. Also, we moved the sentence of comparison with other registries from the result session to discussion, and we deleted some duplicated sentences.

  • The main purpose of the registry is missing as there is no analysis of policies which are the main goal of data collection in this registry. As such I suggest either to describe AMI patients in the registry or show some implementation of policy changes
  • Response)
  • We did not understand your comment exactly. However, since the purpose of registry development is omitted, it is understood that it should be described, so we have been added some purposes of the registry to the introduction. (Line 87-89)

Round 2

Reviewer 2 Report

Thank you for the revision of the manuscript. I have no further remarks.

Reviewer 3 Report

No further comments